# A New Adaptive High-Degree Unscented Kalman Filter with Unknown Process Noise

**Daxing Xu, Bao Wang, Lu Zhang and Zhiqiang Chen ***

College of Electrical and Information Engineering, Quzhou University, Quzhou 324000, China;
dxxu@qzc.edu.cn (D.X.); neil123@qzc.edu.cn (B.W.); zhanglu@qzc.edu.cn (L.Z.)
* Correspondence: czq@qzc.edu.cn

**Abstract:** Vehicle state, including location and motion information, plays an essential role on the Internet of Vehicles (IoV). Accurately obtaining the system state information is the premise of realizing precise control. However, the statistics of system process noise are often unknown due to the complex physical process. It is challenging to estimate the system state when the process noise statistics are unknown. This paper proposes a new adaptive high-degree unscented Kalman filter based on the improved Sage–Husa algorithm. First, the traditional Sage–Husa algorithm is improved using a high-degree unscented transform. A noise estimator suitable for the high-degree unscented Kalman filter is obtained to estimate the statistics of the unknown process noise. Then, an adaptive high-degree unscented Kalman filter is designed to improve the accuracy and stability of the state estimation system. Finally, the target tracking simulation results verify the proposed algorithm's effectiveness.

**Keywords:** nonlinear system; state estimation; unknown system noise; Kalman filter; Sage-Husa; adaptive filter





## 1. Introduction

The automotive industry has experienced decades of rapid growth, and will continue to do so for the foreseeable future. Vehicle state information, including vehicle position, speed, acceleration, orientation, etc., provides a vast opportunity to develop motion-sensing technologies that facilitate the construction of driver assistance systems. Vehicle state estimation is a crucial enabler of the Internet of Vehicles (IoV) [1,2]. To obtain accurate vehicle state information, a significant challenge in vehicle state estimation is noise interference, especially the system's process noise and the sensors' measurement noise. Kalman filtering is essential for dealing with noise disturbances in linear systems. In 1960, Kalman published the famous Kalman filtering method, which marked the establishment of modern filtering theory [3]. The Kalman filter (KF) has been widely used due to the development of numerical calculations. However, the system is often nonlinear in practical engineering applications, especially measurement equations. It is hard to obtain an accurate optimal filtering solution because the filtering of nonlinear systems requires infinite-dimensional integration operations [4].

Bucy and Sunahara proposed an extended Kalman filter (EKF) based on the KF framework and Taylor expansion [5]. This algorithm expands the nonlinear equation of the system by Taylor expansion. Then, it truncates the first-degree linear expansion to achieve the purpose of a linear approximation to the system equation, and finally, performs Kalman filtering. However, the EKF has the disadvantages of poor stability, low accuracy, and slow response to target maneuvering. Julier et al. proposed a new filter estimation algorithm, namely the unscented Kalman filter (UKF) [6]. It calculates the predicted value and measured value of the target by the unscented transform based on the method of sampling points. However, it does not use the traditional way of linearizing the nonlinear function such as the EKF [7,8]. UKF does not ignore the high-degree terms of Taylor expansion linearization as in the extended Kalman filter. In this case, it avoids truncation

errors. Due to the advantages of the UKF, the algorithm is widely used in target tracking, airstrike, and navigation [9,10].

However, the UKF is prone to the curse of dimensionality for high-dimensional system states. It causes the filtering performance to be significantly reduced, and fails to keep up with the target. In the algorithm's iterative process, the matrix's square root operation is needed. If the matrix is non-positive definite, the UKF will no longer apply. The cubature Kalman filter (CKF) uses the spherical integration and radial integration criterion to optimize the sigma point sampling strategy and weight distribution of the UKF. It solves the problem of the dimensionality disaster. The algorithm also improves the filtering accuracy and stability [11,12]. The CKF is a particular case of the UKF when the free parameter equals zero. It provides a strictly theoretical basis for the degree of freedom to be zero in the high-dimensional state estimation [13,14]. Reference [15] proposed a high-degree unscented Kalman filter (HUKF) with the analytical solution based on the fifth-degree cubature transform. Introduced free parameters eliminate the unknown degrees of freedom in the solution of high-degree unscented transformation. Theoretical analysis proves that the HUKF can obtain higher accuracy than above filters. Therefore, this paper uses the HUKF framework to design the filter.

When targeted for the nonlinear filtering problem, the algorithms mentioned above require the statistical characteristics of the system noise [16]. However, such a requirement is often unaffordable in practice. Because the system noise is time-varying and unknown, the direct application of the current filtering algorithms suffers from filtering divergence. When the system noise variance is known or small, the literature [17,18] attempted to use the Sage–Husa noise estimator to estimate the noise variance. However, due to the subtraction operation in the estimation process, it is easy to lose the positive definiteness of the estimated noise variance. Moreover, the weight of the noise is updated using the exponential weighting method, resulting in the update rate of the weight knowledge not changing with the noise change. In order to deal unknown noise statistics, the literature [19] derived the conventional Sage–Husa noise estimation method. It solves the problem of noise estimation for linear systems. The literature [20] pointed out that the Sage–Husa noise estimator is only effective for estimating another unknown noise statistic under the premise that the system noise statistic is known. The literature [21] conducted research on the adaptive UKF algorithm and its application. The literature [22] conducted research on nonlinear methods based on neural networks and Bayesian criteria. Thus, the estimated result is inaccurate when the Sage–Husa noise estimator calculates the system noise statistics. At the same time, the Sage–Husa algorithm cannot be directly embedded in the HUKF algorithm.

To make full use of the superior characteristic meter of HUKF, the UKF algorithm is studied under the unknown system noise statistics. Combined with the high-degree unscented transform rules, this paper derives a noise estimator suitable for nonlinear conditions based on the conventional Sage–Husa method. Furthermore, an adaptive HUKF (AHUKF) algorithm is given to realize the purpose of real-time estimation and the correction of noise statistics. The main contributions of this paper include:

(1) The traditional Sage–Husa algorithm is improved, and an accurate estimation method of the process noise's statistics is given for the nonlinear system.
(2) Based on the improved Sage–Husa algorithm, an AHUKF algorithm is provided, which improves the accuracy and stability of the system state estimate. The simulation example of target tracking illustrates the effectiveness of the proposed algorithm.

The remaining parts of this work are summarized as follows. Section 2 briefly introduces the mathematical system model and the problem description. Section 3 presents the high-degree unscented transform rules. The establishment of the HUKF is presented in Section 4. Numerical simulation is presented in Section 5. Finally, Section 6 concludes this paper.

## 2. Problem Formulation

Consider the discrete nonlinear systems as follows:

$$x_k = f(x_{k-1}) + w_k \tag{1}$$

$$z_k = h(x_k) + v_k \tag{2}$$

where $x_k \in \mathrm{R}^n$, $z_k \in \mathrm{R}^m$, $f$, and $h$ are nonlinear functions, $\{w_k\}$ is an independent Gaussian system noise sequence with unknown mean $q_k$ and variance $Q_k$, and $\{v_k\}$ is an independent Gaussian observation noise sequence with known mean $r_k$ and variance $R_k$.

The KF requires both the system process and observation noise to be Gaussian white noise. At the same time, these noises' statistical properties need to be known. The system noise is complex in practical engineering applications, and its statistical characteristics are challenging to know in real-time. In this case, the traditional filtering algorithm cannot get the system state's optimal estimation. This paper assumes that the statistics of process noise are entirely unknown, i.e., its mean and variance are unknown. Although the filter based on the Sage–Husa algorithm can estimate the unknown system noise characteristics, the estimation results are inaccurate when the system noise characteristics change in real-time. The addressed problem is described as follows:

1. For the real-time change of the system noise mean and variance, how to improve the Sage–Husa algorithm to estimate the noise statistics accurately.
2. How to design a HUKF to accurately estimate the state of a nonlinear system with unknown system noise statistics.

## 3. High-Degree Unscented Transform Rules

For a general Gaussian random variable $x \sim N(\overline{x}, P_x)$, a high-degree unscented transform can match high-degree principal moments of the random vector $x$. Thus, the high-degree unscented transform has higher state estimation accuracy than the second-degree unscented transform. Next, we give the first type of sigma points and weights:

$$\chi_0 = \overline{x}, w_0 = \frac{-2n^2 + (4 - 2n)\kappa^2 + (4\kappa + 4)n}{(n + \kappa)^2 (4 - n)} \tag{3}$$

Then, sigma points and weights for the second type:

$$\begin{cases} \chi_{i_1} = \overline{x} + \sqrt{\frac{(n+\kappa)(4-n)}{(\kappa+2-n)}} P_x e_{i_1} \\ \chi_{i_1+n} = \overline{x} - \sqrt{\frac{(n+\kappa)(4-n)}{(\kappa+2-n)}} P_x e_{i_1} \\ w_1 = \frac{(\kappa+2-n)^2}{2(n+\kappa)^2(4-n)} \end{cases} \tag{4}$$

where $e_{i_1}$ is the $i_1$th unit column vector.

Further, the sigma points and weights for the third type can be given as follows:

$$\begin{cases} \chi_{i_2} = \overline{x} + \sqrt{(n+\kappa)} P_x s_{i_2}^+ \\ \chi_{i_2+0.5n(n-1)} = \overline{x} - \sqrt{(n+\kappa)} P_x s_{i_2}^+ \\ \chi_{i_2+n(n-1)} = \overline{x} + \sqrt{(n+\kappa)} P_x s_{i_2}^- \\ \chi_{i_2+1.5n(n-1)} = \overline{x} - \sqrt{(n+\kappa)} P_x s_{i_2}^- \\ w_2 = \frac{1}{(n+\kappa)^2} \end{cases} \tag{5}$$

where $i_2 = 1, 2, \ldots, 0.5n(n-1)$. $s_{i_2}^+$ and $s_{i_2}^-$ are the sets of points as shown below:

$$\begin{cases} \left\{ s_{i_2}^+ \right\} = \left\{ \sqrt{1/2}(\mathbf{e}_k + \mathbf{e}_l) \right\} \\ \left\{ s_{i_2}^- \right\} = \left\{ \sqrt{1/2}(\mathbf{e}_k - \mathbf{e}_l) \right\} \end{cases} \tag{6}$$

Then, the following algebraic equation about $\kappa$ is obtained:

$$(n-1)\kappa^2 + (2n^2 - 14n)\kappa + n^3 - 13n^2 + 60n - 60 = 0 \tag{7}$$

**Remark 1.** *For two-dimensional and three-dimensional systems, $\kappa$ has an optimal solution. When$\kappa$ takes the optimal value, the accuracy of the high-degree unscented transform is higher than that of the fifth-degree cubature transform and fifth-degree unscented transform. For a four-dimensional system,$\kappa$ can only be set to 2. In this case, the high-degree unscented transform is equivalent to the fifth-degree cubature transform and the fifth-degree unscented transform. For one-dimensional and four-dimensional systems or more, there is no optimal$\kappa$ from the perspective of accuracy, but from the standpoint of numerical stability,$\kappa = 2$ can be set.*

## 4. Adaptive High-Degree Unscented Kalman Filter

### 4.1. Filter Design

We first give an HUKF to facilitate the description of solving the statistical characteristics of noise. The following subsection will provide the mean and variance of the system noise involved in the AHUKF algorithm. The specific steps are as follows:

#### 4.1.1. One-Step Prediction of the State

(1) Assume that $P_{k-1|k-1}$ is known, $S_{k-1|k-1}$ is obtained by Cholesky factorization as follows:

$$P_{k-1|k-1} = S_{k-1|k-1}S_{k-1|k-1}^T \tag{8}$$

where $S_{k-1|k-1}$ is the Cholesky factorization of $P_{k-1|k-1}$.

(2) Calculate the sigma points and their weights for $x_{k-1}$:

$$\chi_{00,k-1|k-1} = \hat{x}_{k-1|k-1} \tag{9}$$

(3) Calculate the second type of sigma points $\chi_{1i_1,k-1|k-1}$, $\chi_{2i_1,k-1|k-1}$ and their weights:

$$\begin{cases} \chi_{1i_1,k-1|k-1} = \hat{x}_{k-1|k-1} + \sqrt{\frac{(n+\kappa)(4-n)}{(\kappa+2-n)}}\mathbf{S}_{k-1|k-1}e_{i_1} \\ \chi_{2i_1,k-1|k-1} = \hat{x}_{k-1|k-1} - \sqrt{\frac{(n+\kappa)(4-n)}{(\kappa+2-n)}}\mathbf{S}_{k-1|k-1}e_{i_1} \end{cases} \tag{10}$$

(4) Calculate the third type of sigma points $\chi_{3i_2,k-1|k-1}$, $\chi_{4i_2,k-1|k-1}$, $\chi_{5i_2,k-1|k-1}$, $\chi_{6i_2,k-1|k-1}$ and their weights:

$$\begin{cases} \chi_{3i_2,k-1|k-1} = \hat{x}_{k-1|k-1} + \sqrt{(n+\kappa)}\mathbf{S}_{k-1|k-1}s_{i_2}^+ \\ \qquad\qquad\cdots \\ \chi_{6i_2,k-1|k-1} = \hat{x}_{k-1|k-1} - \sqrt{(n+\kappa)}\mathbf{S}_{k-1|k-1}s_{i_2}^- \end{cases} \tag{11}$$

(5) Combing the function $f(\cdot)$, propagate the sigma points of $x_{k-1}$ to obtain the following points:

$$\begin{cases} \chi_{00,k|k-1}^* = f(\chi_{00,k-1|k-1}) + \hat{q}_{k-1} \\ \qquad\qquad\cdots \\ \chi_{6i_2,k|k-1}^* = f(\chi_{6i_2,k-1|k-1}) + \hat{q}_{k-1} \end{cases} \tag{12}$$

where $\hat{q}_{k-1}$ is the estimated mean value of system noise, and its calculation will be given in the next section.

(6) Calculate the one-step prediction value $\hat{x}_{k|k-1}$ of the state:

$$\hat{x}_{k|k-1} = w_0\chi_{00,k|k-1}^* + w_1\sum_{i_1=1}^n (\chi_{1i_1,k|k-1}^* + \chi_{2i_1,k|k-1}^*) + w_2\sum_{i_2=1}^{0.5n(n-1)}\sum_{l=3}^6 \chi_{li_2,k|k-1}^* \tag{13}$$

(7)    Calculate $P_{k|k-1}$ of the state:

$$P_{k|k-1} = w_0 \chi^*_{00,k|k-1} \chi^{*T}_{00,k|k-1} + w_1 \sum_{i_1=1}^{n} \left( \chi^*_{1i_1,k|k-1} \chi^{*T}_{1i_1,k|k-1} + \chi^*_{2i_1,k|k-1} \chi^{*T}_{2i_1,k|k-1} \right)$$

$$+ w_2 \sum_{i_2=1}^{0.5n(n-1)} \sum_{l=3}^{6} \chi^*_{li_2,k|k-1} \chi^{*T}_{li_2,k|k-1} - \hat{x}_{k|k-1} \hat{x}^T_{k|k-1} + \hat{Q}_{k-1}$$

(14)

where $\hat{Q}_{k-1}$ is the system noise variance estimated in real time, and its calculation will be given in the next section.

4.1.2. One-Step Prediction of Measurement

(1)    Decompose $P_{k|k-1}$ by Cholesky, and get $S_{k|k-1}$:

$$P_{k|k-1} = S_{k|k-1} S^T_{k|k-1} \tag{15}$$

(2)    Compute the first type of sigma point $\chi_{00,k|k-1}$ of $x_k$:

$$\chi_{00,k|k-1} = \hat{x}_{k|k-1} \tag{16}$$

(3)    Calculate the second type of sigma point $\chi_{1i_1,k|k-1}$ and $\chi_{2i_1,k|k-1}$:

$$\begin{cases} \chi_{1i_1,k|k-1} = \hat{x}_{k|k-1} + \sqrt{\frac{(n+\kappa)(4-n)}{(\kappa+2-n)}} \mathbf{S}_{k|k-1} e_{i_1} \\ \chi_{2i_1,k|k-1} = \hat{x}_{k|k-1} - \sqrt{\frac{(n+\kappa)(4-n)}{(\kappa+2-n)}} \mathbf{S}_{k|k-1} e_{i_1} \end{cases} \tag{17}$$

(4)    Calculate the third type of sigma point $\chi_{3i_2,k|k-1}, \chi_{4i_2,k|k-1}, \chi_{5i_2,k|k-1}$ and $\chi_{6i_2,k|k-1}$:

$$\begin{cases} \chi_{3i_2,k|k-1} = \hat{x}_{k|k-1} + \sqrt{(n+\kappa)} \mathbf{S}_{k|k-1} s^+_{i_2} \\ \quad \cdots \\ \chi_{6i_2,k|k-1} = \hat{x}_{k|k-1} - \sqrt{(n+\kappa)} \mathbf{S}_{k|k-1} s^-_{i_2} \end{cases} \tag{18}$$

(5)    Compute the propagated sigma points of $x_k$:

$$\begin{cases} Z_{00,k|k-1} = f\left( \chi_{00,k-1|k-1} \right) + \hat{r}_k \\ \quad \cdots \\ Z_{6i_2,k|k-1} = f\left( \chi_{6i_2,k-1|k-1} \right) + \hat{r}_k \end{cases} \tag{19}$$

(6)    Calculate $\hat{z}_{k|k-1}$ of the measurement:

$$\hat{z}_{k|k-1} = w_0 Z_{00,k|k-1} + w_1 \sum_{i_1=1}^{n} \left( Z_{1i_1,k|k-1} + Z_{2i_1,k|k-1} \right) +$$

$$w_2 \sum_{i_2=1}^{0.5n(n-1)} \left( Z_{3i_2,k|k-1} + Z_{4i_2,k|k-1} + Z_{5i_2,k|k-1} + Z_{6i_2,k|k-1} \right)$$

(20)

(7)    Calculate $P_{zz,k|k-1}$ of the measurement at time $k$:

$$P_{zz,k|k-1} = w_0 Z_{00,k|k-1} Z^T_{00,k|k-1} + w_1 \sum_{i_1=1}^{n} \left( Z_{1i_1,k|k-1} Z^T_{1i_1,k|k-1} + Z_{2i_1,k|k-1} Z^T_{2i_1,k|k-1} \right) + w_2 \sum_{i_2=1}^{0.5n(n-1)} \sum_{l=3}^{6} Z_{li_2,k|k-1} Z^T_{li_2,k|k-1} - \hat{z}_{k|k-1} \hat{z}^T_{k|k-1} + R_k \tag{21}$$

### 4.1.3. Filter Update

(1)  Calculate the gain $W_k$ of the HUKF at time $k$:

$$W_k = P_{xz,k|k-1} P_{zz,k|k-1}^{-1} \tag{22}$$

(2)  Calculate the estimate of the HUKF at time $k$:

$$\hat{x}_{k|k} = \hat{x}_{k|k-1} + W_k \varepsilon_k \tag{23}$$

(3)  Compute $P_{k|k}$ of the HUKF at time $k$:

$$P_{k|k} = P_{k|k-1} - W_k P_{zz,k|k-1} W_k^T \tag{24}$$

**Remark 2.** *For the systems (1) and (2), given the initial state of the state, the AHUKF can be performed according to the above three steps to obtain a state estimate value, where the mean and variance of the system noise in (12) and (14) are given in the next subsection.*

### 4.2. System Noise Statistics Estimation

Under the case of time-invariant system noise, the system noise average estimation formula based on the Sage–Husa algorithm is:

$$\hat{q}_k = \frac{1}{j} \sum_{k=1}^{j} [\hat{x}_{k|k} - f(\hat{x}_{k-1|k-1})] \tag{25}$$

Then, the estimation of $q_k$ of the system noise under the unscented transformation rule is:

$$\hat{q}_k = \frac{1}{j} \sum_{k=1}^{j} [\hat{x}_{k|k} - \sum_{i=0}^{2} w_i f(\hat{x}_{k-1|k-1})] \tag{26}$$

The recursive formula of the system noise covariance matrix is:

$$\hat{Q}_k = \frac{1}{k}[(k-1)\hat{Q}_{k-1} + W_k \varepsilon_k \varepsilon_k^T W_k^T + P_{k|k} - (\sum_{i_1=0}^{2} w_i(\chi_{i,k-1|k-1}^* \chi_{i,k-1|k-1}^{*T})\hat{x}_{k|k-1}\hat{x}_{k|k-1}^T)] \tag{27}$$

where $\varepsilon_k$ is the innovation sequence, i.e., $\varepsilon_k = z_k - \hat{z}_{k|k-1}$. When the system noise's statistics are unknown, the noise estimator is obtained:

$$\hat{q}_k = (1 - \mu_k)\hat{q}_{k-1} + \mu_k(\hat{x}_{k|k} - \sum_{i=0}^{2} w_i f(\chi_{i,k-1|k-1})) \tag{28}$$

$$\hat{Q}_k = (1 - \mu_k)\hat{Q}_{k-1} + \mu_k[W_k \varepsilon_k \varepsilon_k^T W_k^T + P_{k|k} - (\sum_{i_1=0}^{2} w_i(\chi_{i,k-1|k-1}^* \chi_{i,k-1|k-1}^{*T})\hat{x}_{k|k-1}\hat{x}_{k|k-1}^T)] \tag{29}$$

where $\mu_k = (1 - \vartheta)/(1 - \vartheta^k)$, $\vartheta$ is the forgetting factor. It is selected in the range of $0.95 < \vartheta < 0.99$.

**Remark 3.** *The literature [23] derived the conventional Sage–Husa noise estimation algorithm based on the Kalman filter framework and used it to solve the filtering problem of unknown noise statistics under linear conditions. However, this algorithm is no longer applicable when the system is nonlinear. We estimate the statistics of the system noise mean (26) and variance (29) in real-time by introducing the unscented transformation rules, and finally, bring the estimation results into Equations (12) and (14).*

**Remark 4.** *The HUKF calculation steps are shown in Equations (3)–(24). Since the filter gain K calculation includes the matrix inversion operation, the calculation amount is much more significant than other equations. Compared with the HUKF, the calculation of AHUKF has more Equations (25)–(29). The computation of these equations only includes multiplication and addition operations. Therefore, AHUKF is only slightly more computationally intensive than HUKF.*

## 5. Result and Discussion

In this section, a two-dimensional space is selected. The target state is a four-dimensional vector $x(k) = \left[ x_k \, v_k^x \, y_k \, v_k^y \right]^T$, where $x_k$ and $y_k$ are the displacements due east and due north, $v_k^x$ represents the velocity component of the target in the true east direction, and $v_k^y$ represents the velocity component of the target in the true north direction, and the sampling period is $T = 1$. The target motion equation is:

$$x_k = F_{k-1}x_{k-1} + w_{k-1} \tag{30}$$

where $F_{k-1} = \begin{bmatrix} 1 & T & 0 & 0 \\ 0 & 1 & 0 & 0 \\ 0 & 0 & 1 & T \\ 0 & 0 & 0 & 1 \end{bmatrix}$.

The observation equation is:

$$z(k) = \begin{bmatrix} \sqrt{x^2(k)+y^2(k)} \\ \arctan\left[\frac{y(k)}{x(k)}\right] \end{bmatrix} + v_k \tag{31}$$

The system noise's statistics are unknown. In order to verify the conclusion, it is set to mean zero, and the covariance matrix is:

$$Q_k = \begin{bmatrix} \frac{aT^3}{3} & \frac{aT^2}{2} & 0 & 0 \\ \frac{aT^2}{2} & aT & 0 & 0 \\ 0 & 0 & \frac{aT^3}{3} & \frac{aT^2}{2} \\ 0 & 0 & \frac{aT^2}{2} & aT \end{bmatrix} \tag{32}$$

The a priori system noise parameter $a = 0.1$, and the actual system noise parameters change in the following three stages:

$$a = \begin{cases} 1 & 1 \le k \le 40 \\ 4 & 41 \le k \le 70 \\ 10 & 71 \le k \le 100 \end{cases} \tag{33}$$

The covariance $R_k$ of the measurement noise $R_k = [0.15 \ 0.01; \ 0.01 \ 0.01]$. Root mean square error (RMSE) is defined as:

$$E_{\text{RMSE}} = \sqrt{\frac{1}{N}\sum_{k=1}^{N}(x_k - \hat{x}_k)^2} \tag{34}$$

where $N$ is the number of simulations.

The simulation results are shown from Figures 1–5 and from Tables 1–3, respectively.

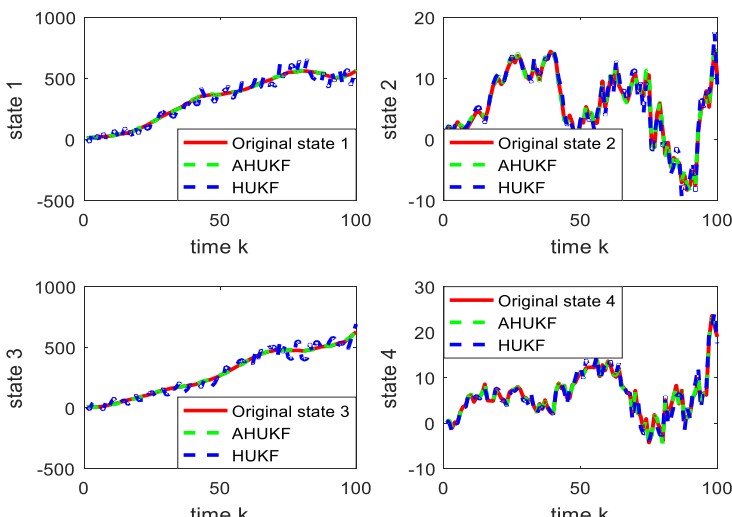

**Figure 1.** True trajectory and estimated trajectory for four states.

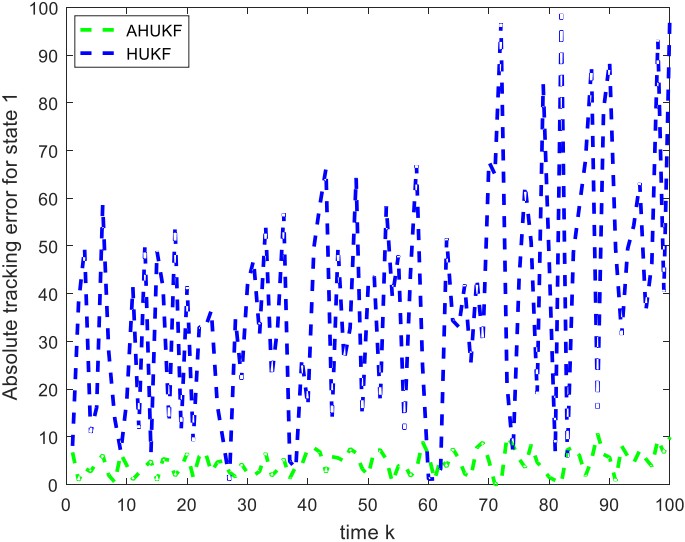

**Figure 2.** Tracking error curve for State 1.

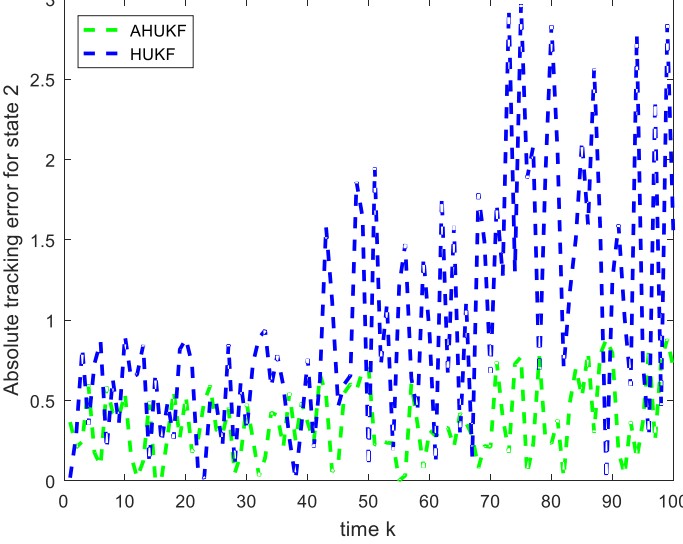

**Figure 3.** Tracking error curve for State 2.

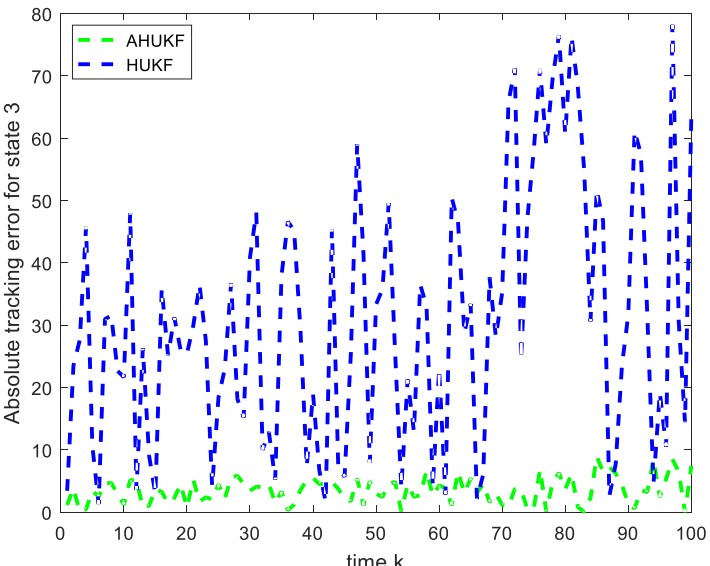

**Figure 4.** Tracking error curve for State 3.

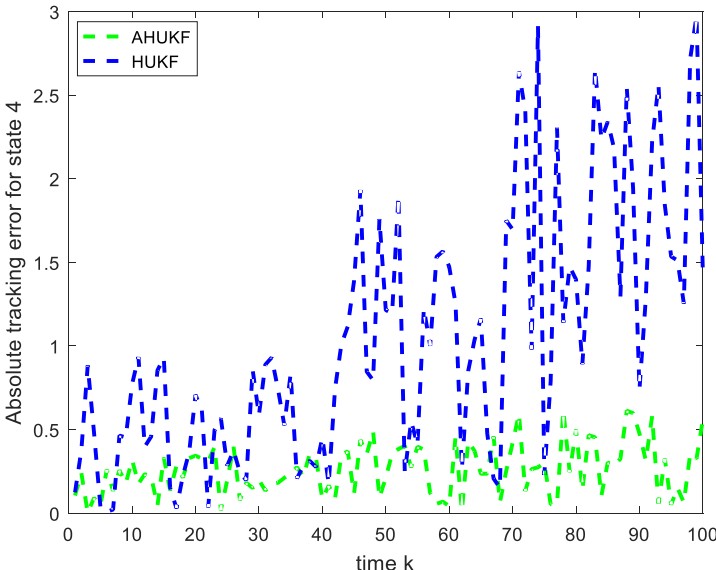

**Figure 5.** Tracking error curve for State 4.

**Table 1.** Comparison of RMSE for four states.

| Algorithm | $x_k$ | $v^x_k$ | $y_k$ | $v^y_k$ |
|-----------|-------|---------|-------|---------|
| AHUKF | 5.2424 | 0.4358 | 4.0745 | 0.3021 |
| HUKF | 45.2371 | 1.2218 | 37.0694 | 1.2829 |

**Table 2.** RMSE with AHUKF at different stages.

| Stages | $x_k$ | $v^x_k$ | $y_k$ | $v^y_k$ |
|--------|-------|---------|-------|---------|
| 1–40 times | 4.1951 | 0.3598 | 3.5098 | 0.2334 |
| 41–70 times | 5.5143 | 0.3820 | 3.8360 | 0.3080 |
| 71–100 times | 6.1430 | 0.5607 | 4.9192 | 0.3697 |

**Table 3.** RMSE with HUKF at different stages.

| Stages | $x_k$ | ${v^x}_k$ | $y_k$ | ${v^y}_k$ |
|---|---|---|---|---|
| 1–40 times | 32.3877 | 0.5904 | 27.7310 | 0.5418 |
| 41–70 times | 41.6727 | 1.1136 | 30.2155 | 0.8616 |
| 71–100 times | 60.7130 | 2.0668 | 51.4019 | 1.9354 |

Both algorithms can track the target trajectory from the state tracking curve in Figure 1. However, from Figures 2–5, the absolute tracking error value of the four states with AHUKF is much smaller than that of HUKF algorithm. In general, the absolute error curve will rise as the noise increases. The error curve of HUKF increases obviously with the growth of noise, whereas the estimation error of AHUKF increases slowly with the more extensive system noise. The result is that AHUKF can estimate noise in real time, but HUKF cannot. According to the statistical results of Table 1, RMSE for the position of the AHUKF algorithm is about one-tenth of the HUKF algorithm, and RMSE for speed is about one-eighth of the latter.

From the statistical results of RMSE at each time stage in Tables 2 and 3, the RMSE value of AHUKF for the four states changes slowly. On the contrary, the RMSE of HUKF to the state becomes significantly large as the noise changes. In particular, in the 71–100 time period, the RMSE increases faster than in the 41–70 time period. The result is that HUKF cannot track the noise statistics if the noise statistical characteristics are unknown. At the same time, AHUKF has a better estimation performance for unknown noise.

## 6. Conclusions

This paper uses a high-degree unscented transform to improve the traditional Sage–Husa algorithm with unknown system noise statistics. Then, the real-time and accurate estimation of the unknown system noise statistics is realized. Furthermore, an AHUKF algorithm based on an improved Sage–Husa algorithm is given. According to the simulation results, it is concluded that the proposed algorithm can effectively overcome the shortcomings of low filtering accuracy and the divergence of traditional nonlinear algorithms where the system noise is unknown. The proposed algorithm also improves the adaptability and stability of the filter. However, system noise is non-Gaussian in some practical applications. In the case of non-Gaussian noise, solving the state estimation problem of nonlinear systems is particularly important. Therefore, we will study the filtering problem of nonlinear non-Gaussian systems in future work.

**Author Contributions:** Methodology, D.X.; software, B.W.; formal analysis, D.X., L.Z. and Z.C.; investigation, D.X.; resources, L.Z. and Z.C.; data curation, B.W.; writing—D.X. and B.W.; writing—review and editing, L.Z. and Z.C.; project administration, B.W.; funding acquisition, L.Z. and Z.C. All authors have read and agreed to the published version of the manuscript.

**Funding:** This work was supported by the Natural Science Foundation of Zhejiang Province under Grant LZY22E050003, LZY22E050005; and the Zhejiang Public Welfare Project under Grant LGN20C050002.

**Acknowledgments:** The authors would like to thank the anonymous reviewers for their constructive comments and suggestions, which strengthened this paper a lot.

**Conflicts of Interest:** The authors declare no conflict of interest.

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
