# Peer review of "A New Adaptive High-Degree Unscented Kalman Filter with Unknown Process Noise"

_electronics, doi:10.3390/electronics11121863_

Round 1

Reviewer 1 Report

The statistical characteristics of system noise are supposed to be unknown, however, what kind of restriction should be stablished instead (bounds, kind of noise, noise/signal relation bound, etc)?

The shown comparison  in figures 2-5 are related to  the algorithm “High-degree unscented Kalman filter” (HUKF) vs an adaptive high-degree unscented Kalman filter (AHUKF), i.e. the HUKF vs the noise- adaptive version. The comparison seems not to be fair. What can said the authors with respect to this?. 

On page 11, lines 239-240 the authors claims:  "This result is that AHUKF can estimate noise in real time, but HUKF cannot."  The objective of the HUKF is not he noise estimation, more over nothing is said about the complexity or about the number of float point operatios of each algorithm (in order to make the comparison more complete).

Author Response

Dear reviewer, please see attachment for detailed response.

Reviewer 2 Report

This paper studies an adaptive high-degree unscented Kalman filter based on the improved Sage-Husa algorithm. First, the traditional Sage-Husa algorithm is improved using a high-degree unscented transform. A noise estimator suitable for the high-degree unscented Kalman filter is obtained to estimate the statistical characteristics of the unknown process noise. Then, an adaptive high-degree unscented Kalman filter is designed to improve the accuracy and stability of the state estimation system. Finally, the simulation results of the target tracking verify the effectiveness of the proposed algorithm. The first problem I noticed is the presentation. The structure is not easy to follow. There are many grammatical errors. I suggest a professional editing service should be used.

The novelty and contribution of the paper are not sufficiently highlighted in the introduction. The main difficulties of the paper should be clearly set out against the existing works. I am not sure if this is done properly given the current writing.

Gaussian noise is not defined here. The model is somehow out of the blue without sufficient justifications. 

The meaning of the algebriac equation (7) should be explained. Why is it useful in reality?

A major issue is the equation (14). A term is missing in the second line, which invalidates the process. Hopefully this is only an omission in the presentation. Please double check the process and ensure the analysis and experiments below are valid.

In the conclusion, it is mentioned that The simulation results show that the proposed algorithm can effectively overcome the shortcomings of low filtering accuracy and divergence of traditional nonlinear algorithms when the statistical characteristics of system noise are unknown and time-varying. The proposed algorithm also improves the adaptability and stability of the filter. This is not sufficiently supported by the evidence provided in the paper. 

Author Response

(The authors gave the same response as above.)

Round 2

Reviewer 2 Report

The revised paper is a pleasant read. I recommend it for publication.